# Correlating Variational Autoencoders Natively For Multi-View Imputation

**Ella S. C. Orme**
Department of Mathematics
Imperial College London
ella.orme18@imperial.ac.uk

**Marina Evangelou**
Department of Mathematics
Imperial College London
m.evangelou@imperial.ac.uk

**Ulrich Paquet**
African Institute for Mathematical Sciences, South Africa
ulrich@aims.ac.za

## Abstract

Multi-view data from the same source often exhibit correlation. This is mirrored in correlation between the latent spaces of separate variational autoencoders (VAEs) trained on each data-view. A multi-view VAE approach is proposed that incorporates a joint prior with a non-zero correlation structure between the latent spaces of the VAEs. By enforcing such correlation structure, more strongly correlated latent spaces are uncovered. Using conditional distributions to move between these latent spaces, missing views can be imputed and used for downstream analysis. Learning this correlation structure involves maintaining validity of the prior distribution, as well as a successful parameterization that allows end-to-end learning.

## 1 Introduction

Data from multiple sources describing the same subjects arises in a wealth of settings. This can be clinical information of patients alongside genetic information and scan data. Datasets consisting of multiple views are referred to as multi-view or multi-modal data. There are instances where not all views are always available for every realisation. For example, a patient may miss an appointment or machinery may falter, resulting in no reading for a specific view. Missing data results in smaller usable datasets and reduced statistical power with many methods only applicable to full datasets [1] and can result in reduced performance in downstream analysis [2]. This manuscript presents a multi-view imputation approach, where its aim is to impute the realisations of a missing view by incorporating the information learnt from the other view.

The proposed multi-view imputation method, named Joint Prior Variational Autoencoder (JPVAE), is based on variational autoencoders (VAEs) [3], with the views connected solely through a joint prior on the VAEs' latent embeddings. Standard autoencoders seek to encode a latent representation of data, and from this encoding reconstruct the original data via a decoder. Multi-view approaches allow the latent representation of a missing view to be obtained, from which the reconstruction can be used as an imputation of the missing view. Several multi-view imputation approaches exist in the literature based on autoencoders [4]. However, as variational autoencoders learn a continuous embedding, they provide better interpolation of the latent space than standard autoencoders, making them a more suitable approach for an imputation method. The proposed joint prior in JPVAE incorporates a non-zero correlation structure that is found to increase the observed correlation between views in the latent spaces. This allows for successful movement between latent spaces, improving the ability to impute missing views.

Preprint.

Various approaches have been proposed extending VAEs to the multi view case, including those by [5–8], with missing data imputation included as a feature of these methods. Most recently, proposed methods differ via their method of approximating a joint posterior. Daunhawer et al. [9] discuss the undesirable upper bound these methods put on a lower limit of the log-likelihood used as the objective function in VAEs. In contrast to existing approaches, which largely assume some joint posterior and/or shared latent space, our work has the novelty that is based on a joint prior between the latent variables. To the best of our knowledge, this is the first attempt made to correlate the latent spaces of VAEs natively through a joint prior.

JPVAE is most similar in structure to the private version of Deep Variational Canonical Correlation Analysis (VCCA-private), a multi-view VAE approach by Wang et al. [10]. However their model contains both private and shared variables in the latent space and doesn't have a suitable approach for imputing missing views. JPVAE can also be seen as an application of the ideas from self-supervised learning techniques such as Barlow Twins [11], where the embeddings from noisy versions of the same input are driven to be highly correlated via a loss function.

Figure 1 illustrates an application of the proposed approach JPVAE, on imputing view 2 of the data that corresponds to the bottom half of MNIST digits using the top half of the digits data (view 1). Through the proposed approach, reconstructions of missing views can be obtained. This is achieved through the conditional distribution between the two latent variables, as illustrated in Figure 2. By incorporating a joint prior with a non-zero cross-correlation structure, we observe better quality results in imputing view 2. Further realisations of image imputation, including imputation of view 1 from view 2, can be found in Appendix A.5.

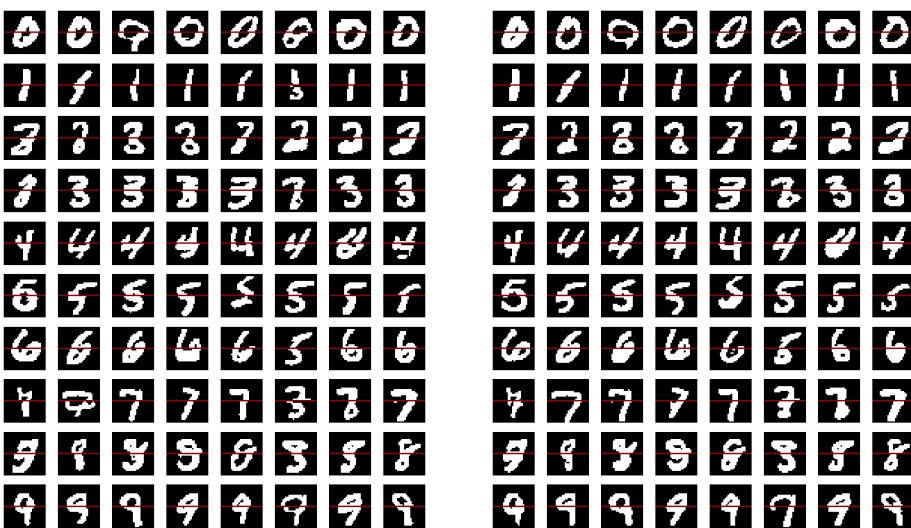

(a) No correlation learnt, but estimated empirically after training

(b) Correlation learnt natively

Figure 1: Imputation of the bottom half of MNIST digits (view 2 of the data) using the top half of the image (view 1) on a JPVAE model trained with (a) independent priors (completely separate VAEs) and (b) a joint prior with learnt correlation structure between latent spaces. The cross entropy loss between true bottom half of image and imputation is 109.1 in (a) and 93.04 in (b). A classifier trained on the concatenation of the reconstruction of view 1 and imputation of view 2 achieves average testing accuracy of 79.92% (1.0) in (a) and 87.45% (0.27) in (b). See Appendix A.5 for further details.

## 1.1 Contributions

A novel multi-view imputation approach based on variational autoencoders, named JPVAE, is proposed that in contrast to existing work assumes a joint prior between the latent variables.[1] As the multiple views are correlated, the generated latent spaces are found also to be correlated. JPVAE

---

[1]The code for implementation of JPVAE and the numerical experiments contained in this manuscript can be found at https://github.com/eso28599/JPVAE.

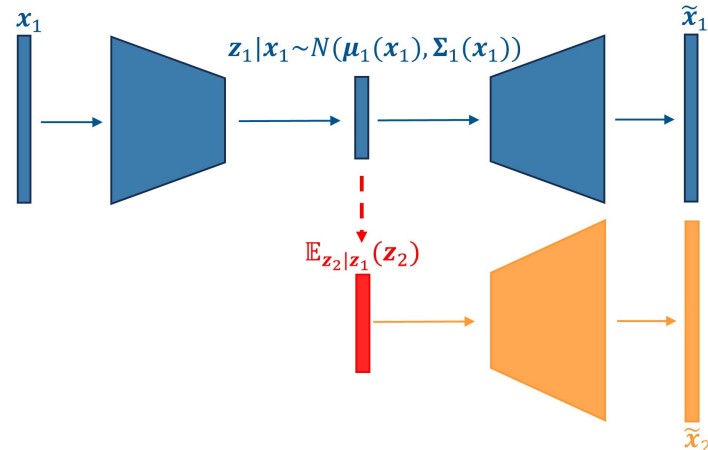

Figure 2: Workflow to obtain reconstruction $\tilde{x}_1$ and imputation $\tilde{x}_{2|1}$ from input $x_1$ only, on pre-trained JPVAE network. Trapeziums represent encoders/decoders and rectangles represent vectors. The structures for view 1 and 2 are shown in blue and orange respectively. The expectation step is shown in red.

takes advantage of this correlation and whilst marginals for each VAE are assumed to follow a standard normal distribution, a joint prior is assumed with a non-zero cross correlation. The imposed correlation structure is learnt natively by forcing a stronger correlation in the latent space. Section 2 presents the proposed method and related theorems that showcase the validity of the proposed method. As part of our work, we present and prove theorems that enable the parameterization of differentiable positive definite matrices that allow end-to-end learning. Through our conducted experiments presented in Section 3, learning the correlation structure leads to improved imputation ability, lower reconstruction loss and better predictive likelihood. Imputed views from JPVAE can be used for downstream tasks, with improved classification accuracy demonstrated. Lastly, better VAE models are learnt with JPVAE preventing posterior collapse, a common problem observed with VAEs [12].

### 1.2 Notation

Throughout this paper matrices are denoted by capital letters and column vectors are denoted by lower case letters, both emboldened. $\boldsymbol{X} = \{\mathbf{x}_i\}_{i=1}^{n}$ with vector $\mathbf{x}_i \in \mathbb{R}^c$ represents an entire dataset in $\mathbb{R}^{n \times c}$, with $\mathbf{x}_i$ an individual realisation. A diagonal matrix with vector $\boldsymbol{a}$ along the diagonal is represented by $\mathrm{diag}(\boldsymbol{a})$. A block diagonal matrix with matrices $\boldsymbol{A}_i$ along the diagonal is represented by $\mathrm{bdiag}(\boldsymbol{A}_i)$. Vertical concatenation of column vectors $\boldsymbol{a}$ and $\boldsymbol{b}$ is denoted by $(\boldsymbol{a}; \boldsymbol{b})$ i.e. $(\boldsymbol{a}; \boldsymbol{b}) = (\boldsymbol{a}^T; \boldsymbol{b}^T)^T$. The eigenvalues and singular values of matrix $\boldsymbol{A}$ are denoted by $\lambda_i(\boldsymbol{A})$ and $\sigma_i(\boldsymbol{A})$ respectively, indexed by subscript $i$ in order of decreasing magnitude. $\mathbf{I}$ denotes an identity matrix of relevant dimension.

## 2 Multi-view variational autoenconder with a joint prior

This section presents the novel multi-view VAE approach JPVAE where each view has a separate associated VAE. Before discussing the proposed approach in detail, the main concepts of VAEs are introduced. Subsequently the single-view VAE is extended to the multi-view setting and the joint prior is presented. The different components of JPVAE are presented including theoretical work on matrix parameterization which enables end-to-end learning.

### 2.1 Variational autoencoder

A standard VAE seeks to encode data $\boldsymbol{X}$ in a probabilistic latent space and decode from this space to reconstructed data $\tilde{\boldsymbol{X}}$ [3]. The goal is for $\tilde{\boldsymbol{X}}$ to be as close as possible to $\boldsymbol{X}$. As the latent space is a probabilistic embedding rather than one obtained by a deterministic mapping, it has the advantage that

it can be explored fully, including the sampling of new realisations from the same distribution as $\tilde{\boldsymbol{X}}$. These realisations are assumed to be taken from the same underlying distribution as $\boldsymbol{X}$, $p_\theta(\cdot) = p(\cdot|\theta)$ where $\theta$ is the set of parameters defining the distribution.

VAEs are a type of variational Bayesian method that seek to find a lower bound for this marginal probability, $p_\theta(\mathbf{X})$, through a Bayesian framework. The likelihood of $\boldsymbol{x} \in \mathbb{R}^c$ given latent variable $\boldsymbol{z} \in \mathbb{R}^d$ is denoted by $p_\theta(\cdot|\boldsymbol{z})$ and the prior for the latent variables, usually taken to be a standard normal, is denoted by $p_\theta(\boldsymbol{z})$. The encoder and decoder are neural networks that seek to learn the posterior distribution of $\boldsymbol{z}$ given $\boldsymbol{x}$, $p_\theta(\cdot|\boldsymbol{x})$, and the likelihood of $\boldsymbol{x}$ given $\boldsymbol{z}$ respectively, given the assumed prior over the latent space. In order to make the posterior learnable, an *approximate* posterior distribution is used that usually is a multivariate normal distribution. The approximation $p_\theta(\cdot|\boldsymbol{x}) \approx q_\phi(\cdot|\boldsymbol{x})$ is made where $\phi$ denotes the parameters of the probabilistic encoder, $q_\phi(\cdot|\boldsymbol{x})$. The encoder maps an input $\boldsymbol{x}$ to a mean vector $\boldsymbol{\mu}(\mathbf{x}) \in \mathbb{R}^d$ and to the log of the vector of variances, $\boldsymbol{\sigma}^2(\mathbf{x}) \in \mathbb{R}^d$. A sample is drawn from the approximate posterior distribution and this is used as an input to the neural network which acts as the decoder, $D_\theta$. As $\phi$ appears within the distribution of the latent variables, derivatives cannot simply be taken inside the expectation term. Differentiability of the loss function is required for the loss-driven parameter update steps and therefore a reparamaterization trick needs to be implemented [3].

Instead of drawing $\boldsymbol{z}$ directly, $\boldsymbol{\epsilon} \sim N(\mathbf{0}, \boldsymbol{I})$ is drawn and $\boldsymbol{z} = \boldsymbol{\mu}(\mathbf{x}) + \mathrm{diag}(\boldsymbol{\sigma}^2(\mathbf{x}))\boldsymbol{\epsilon}$ is determined. The distribution over which we take expectation is now independent of the parameters for which we take derivatives, and derivative update steps may now be implemented. The neural network decoder then seeks to reconstruct $\boldsymbol{x}$ from the latent variable $\boldsymbol{z}$.

A maximum likelihood principle is then applied to the marginal likelihood of the data $p_\theta(\boldsymbol{X})$ to obtain estimates for the parameters within the encoders and decoders. As the likelihood is intractable a lower bound on the log-likelihood, known as the Evidence Lower Bound (ELBO) is instead maximised, given by:

$$\mathcal{L}(\theta, \phi) = \mathbb{E}_{\boldsymbol{z} \sim q_\phi(\cdot|\boldsymbol{x})} \left[ \ln(p_\theta(\boldsymbol{x}|\boldsymbol{z})) \right] - D_{KL}(q_\phi(\cdot|\boldsymbol{x})||p_\theta(\cdot)) \tag{1}$$

where $D_{KL}(r|s)$ denotes the Kullback-Leibler (KL) divergence between distributions $r$ and $s$ [13]. $\mathbb{E}_{\boldsymbol{z} \sim q_\phi(\cdot|\boldsymbol{x})}(\cdot)$ denotes the expectation with respect to the conditional distribution of $\boldsymbol{z}$ given $\boldsymbol{x}$. Maximising the ELBO is equivalent to finding a balance between an approximate posterior that is close to the prior and putting weight on the latent variable space that maximises the likelihood of the data given these same variables, $\ln(p_\theta(\boldsymbol{x}|\boldsymbol{z}))$. Without the KL term, the delta function is returned as the approximate posterior and the autoencoder is recovered.

With this VAE formulation the KL term often becomes very small or 'vanishes' leading to the posterior equalling the prior (posterior collapse). This is referred to as KL vanishing and leads to a decoder that is largely independent of the latent variables. See [14] for further discussion of this problem. To combat this, we implement KL annealing – a procedure where a weight $\beta$ is introduced on the KL term and gradually increased, typically from 0, as learning occurs [15]. The objective function becomes:

$$\mathcal{L}_\beta(\theta, \phi) = \mathbb{E}_{\boldsymbol{z} \sim q_\phi(\cdot|\boldsymbol{x})} \left[ \ln(p_\theta(\boldsymbol{x}|\boldsymbol{z})) \right] - \beta D_{KL}(q_\phi(\cdot|\boldsymbol{x})||p_\theta(\cdot)). \tag{2}$$

It is now assumed two data views with shared row dimension are present and represented by $\boldsymbol{X} = \{\boldsymbol{X}_1, \boldsymbol{X}_2\}$ with $\boldsymbol{X}_i \in \mathbb{R}^{N \times c_i}$. For brevity, the sample subscript is dropped and $\boldsymbol{x}_i \in \mathbb{R}^{c_i}$ refers to an individual realisation from view $i$. Here $N$ is the number of rows in both views and $c_i$ is the number of columns in view $i = 1, 2$. This shared row dimension implies the views are paired, with row $j$ representing the same individual/sample in both views. This allows for meaningful correlation in the latent space. The notation introduced in this section for single view VAEs is used in the following sections where multi-view VAEs are presented. Subscripts are used to denote the relevant views e.g. the encoder and decoder in view $i$ are represented by $E_{i\phi_i}$ and $D_{i\theta_i}$ respectively.

## 2.2 Joint prior variational autoencoder

As the different views in a multi-view dataset are from a common source, correlation exists between the views. This translates to correlation between the latent spaces of each view from independently trained VAEs. JPVAE takes advantage of this correlation, enforcing the relationship between the two views via a joint prior on the latent variables, as illustrated in Figure 3. Enforcing the proposed correlation structure between the latent spaces ensures we can move from the original space where

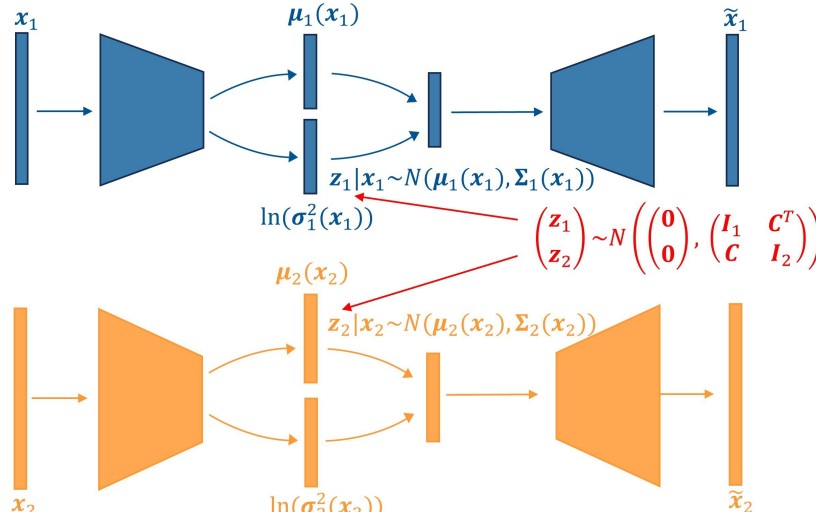

Figure 3: Workflow to obtain reconstructions $\tilde{x}_1$ and $\tilde{x}_2$ from realisations $x_1$ and $x_2$ using a learnt JPVAE structure. Trapeziums present the encoders and decoders of the two views. Vectors are presented by the rectangles. The structures for view 1 and 2 are shown in blue and orange respectively. The prior on the latent variables is shown in red.

data are highly correlated in a non-linear fashion, to a space where the correlation is linear, and back to the reconstructed space that has a non-linear correlation. The marginal prior on the latent variables corresponding to each view is a standard normal, as with the traditional VAE. However, a cross-covariance matrix $C$ is assumed between the latent variables $z_1$ and $z_2$. Having separate VAE structures for each view assumes conditional independence in both directions: (a) given the data the latent variables are independent, and (b) given the latent variables the data are independent. This allows the unique features of each view to be encoded and decoded.

As the latent spaces are linearly correlated, it is possible to move between them via the conditional distribution, as illustrated in Figure 2. This allows for imputation of missing views, obtaining a reconstruction of $x_2$ solely from realisation $x_1$ (and vice versa). If a joint prior is not assumed, separate VAEs are trained on each view and there is no correlation enforced between latent spaces. Whilst some correlation exists between the latent spaces, it is not as strong, and therefore may not produce as accurate reproductions, as illustrated in the numerical experiments of Section 3.

## 2.3 Objective function

By assuming that latent variables for each view are independent given the data, the approximated posterior distributed can be expressed as:

$$q_\phi(z|x) = q_{(\phi_1,\phi_2)}((z_1, z_2)|(x_1, x_2)) = q_{\phi_1}(z_1|x_1)q_{\phi_2}(z_2|x_2). \tag{3}$$

The individual approximate posteriors $q_{\phi_i}(\cdot|x_i)$ are multivariate Gaussians with mean and covariance matrix determined by the output of $E_{i\phi_i}$. For input $x_i$, these are $\mu_i(x_i)$ and $\text{diag}(\sigma_i^2(x_i))$ respectively. As the posteriors are assumed to be independent, the joint distribution is multivariate Gaussian with mean $(\mu_1(x_1); \mu_2(x_2))$. The covariance matrix is represented by $\text{bdiag}(\Sigma_i(x_i))$ with $\Sigma_i(x_i) = \text{diag}(\sigma_i^2(x_i))$. Similarly, by assuming the data is independent given the latent variables, the likelihood function can be expressed as:

$$p_\theta(x|z) = p_{(\theta_1,\theta_2)}((x_1, x_2)|(z_1, z_2)) = p_{\theta_1}(x_1|z_1)p_{\theta_2}(x_2|z_2). \tag{4}$$

This is equivalent to having separate encoders and decoders for each view but with a joint prior. Assuming independence means the expectation term in the ELBO can be separated into terms corresponding to the separate views:

$$\mathbb{E}_{z\sim q_\phi(\cdot|x)}\left[\ln p_\theta(x|z)\right] = \mathbb{E}_{z_1\sim q_{\phi_1}(\cdot|x_1)}\left[\ln p_{\theta_1}(x_1|z_1)\right] + \mathbb{E}_{z_2\sim q_{\phi_2}(\cdot|x_2)}\left[\ln p_{\theta_2}(x_2|z_2)\right].$$

The objective function therefore becomes:

$$\mathcal{L}_\beta(\theta_1, \phi_1, \theta_2, \phi_2) = \mathbb{E}_{\boldsymbol{z}_1 \sim q_{\phi_1}(\cdot|\boldsymbol{x})} \left[\ln p_{\theta_1}(\boldsymbol{x}_1|\boldsymbol{z}_1)\right] + \mathbb{E}_{\boldsymbol{z}_1 \sim q_{\phi_2}(\cdot|\boldsymbol{x})} \left[\ln p_{\theta_2}(\boldsymbol{x}_2|\boldsymbol{z}_2)\right] \tag{5}$$
$$- \beta D_{KL}(q_{\phi_1}(\cdot|\boldsymbol{x}_1)q_{\phi_2}(\cdot|\boldsymbol{x}_2) \parallel p_\theta(\cdot)).$$

## 2.4 Joint prior

There are assumed to be $n_i$ latent variables in the VAE for view $i$, represented by the random vector $\boldsymbol{z}_i \in \mathbb{R}^{n_i}$. Without loss of generality it is that assumed $n_1 \leq n_2$. The joint prior distribution of the random vector $\boldsymbol{z} = (\boldsymbol{z}_1, \boldsymbol{z}_2) \in \mathbb{R}^{n_1+n_2}$ is assumed to be a multivariate normal with mean $\boldsymbol{0}$ and covariance matrix, $\boldsymbol{\Sigma}_C$:

$$\boldsymbol{\Sigma}_C = \begin{pmatrix} \boldsymbol{I}_1 & \boldsymbol{C}^T \\ \boldsymbol{C} & \boldsymbol{I}_2 \end{pmatrix} \tag{6}$$

where $\boldsymbol{C} \in \mathbb{R}^{n_2 \times n_1}$ is the cross-covariance matrix encapsulating the relationship between the two latent spaces. $\boldsymbol{I}_i$ is the $n_i \times n_i$ identity matrix. As $\boldsymbol{\Sigma}_C$ is a covariance matrix, it needs to satisfy two conditions: symmetry and positive semi-definiteness. With the defined structure it is clearly symmetric and so it is sufficient to require $\boldsymbol{\Sigma}_C$ to be a positive semi-definite matrix to ensure it is a well-defined covariance matrix. For a covariance matrix with structure as defined in Eq. 6, it is useful to note that $\boldsymbol{\Sigma}_C$ is positive definite if and only if $\boldsymbol{I}_1 - \boldsymbol{C}^T\boldsymbol{C}$ (or $\boldsymbol{I}_2 - \boldsymbol{C}\boldsymbol{C}^T$) is positive definite [16]. Additionally, as (a) a real symmetric matrix is positive (semi) definite if and only if all of its eigenvalues are positive (non-negative) [17, p. 51] and (b) a matrix is invertible if and only if all of its eigenvalues are non-zero, covariance matrix $\boldsymbol{\Sigma}_C$ (and $\boldsymbol{I}_1 - \boldsymbol{C}^T\boldsymbol{C}/\boldsymbol{I}_2 - \boldsymbol{C}\boldsymbol{C}^T$) is positive definite if and only if it is invertible.

## 2.5 Kullback–Leibler divergence term

As discussed earlier, for calculating the ELBO for the proposed prior, the KL divergence between $q_{\phi_1}(\cdot|\boldsymbol{x}_1)q_{\phi_2}(\cdot|\boldsymbol{x}_2)$ and the prior on $(\boldsymbol{z}_1, \boldsymbol{z}_2)$, $p_C$, is required. The following general result on KL divergence between multivariate Gaussians is applied in this setting.

Between two $k$-dimensional multivariate Gaussians $r$ and $s$ with respective means $\boldsymbol{\mu}_r$ and $\boldsymbol{\mu}_s$ and respective covariance matrices $\boldsymbol{\Sigma}_r$ and $\boldsymbol{\Sigma}_s$, the KL divergence is given by [18]:

$$D_{KL}(s||r) = \frac{1}{2}\left[\ln\frac{|\boldsymbol{\Sigma}_r|}{|\boldsymbol{\Sigma}_s|} - k + (\boldsymbol{\mu}_s - \boldsymbol{\mu}_r)^T\boldsymbol{\Sigma}_r^{-1}(\boldsymbol{\mu}_s - \boldsymbol{\mu}_r) + \text{tr}\left\{\boldsymbol{\Sigma}_r^{-1}\boldsymbol{\Sigma}_s\right\}\right]. \tag{7}$$

As this assumes the existence of $\boldsymbol{\Sigma}_r^{-1}$ and $\boldsymbol{\Sigma}_s^{-1}$, both covariance matrices must be positive definite.

By utilising the result of Eq. 7 with the prior over $\boldsymbol{z} = (\boldsymbol{z}_1, \boldsymbol{z}_2)$ set as $r = p_C$, and the approximate posterior distribution as $s = q_{\phi_1}(\cdot|\boldsymbol{x}_1)q_{\phi_2}(\cdot|\boldsymbol{x}_2)$, the KL term of Eq. 2 is obtained. Explicitly, $p_C = N(\boldsymbol{0}, \boldsymbol{\Sigma}_C)$ and $q_{\phi_1}(\cdot|\boldsymbol{x}_1)q_{\phi_2}(\cdot|\boldsymbol{x_2}) = N((\boldsymbol{\mu}_1(\boldsymbol{x}_1), \boldsymbol{\mu}_2(\boldsymbol{x}_2)), \boldsymbol{\Sigma}_q)$ with $\boldsymbol{\Sigma}_q = \text{bdiag}(\boldsymbol{\Sigma}_i)$. For the utilisation of Eq. 7, it is assumed that $\boldsymbol{\Sigma}_C$ is positive definite and the variances of the approximate posterior are positive.

Using the block matrix inverse result from [19, p. 18] (stated in Appendix A.1.1) the inverse of $\boldsymbol{\Sigma}_C$ is given by:

$$\boldsymbol{\Sigma}_C^{-1} = \begin{pmatrix} \boldsymbol{D}_1 & -\boldsymbol{D}_1\boldsymbol{C}^T \\ -\boldsymbol{D}_2\boldsymbol{C} & \boldsymbol{D}_2 \end{pmatrix} \tag{8}$$

where $\boldsymbol{D}_1 = (\boldsymbol{I}_1 - \boldsymbol{C}^T\boldsymbol{C})^{-1}$ and $\boldsymbol{D}_2 = (\boldsymbol{I}_2 - \boldsymbol{C}\boldsymbol{C}^T)^{-1}$. These inverses are guaranteed to exist, given the assumption of positive definiteness on $\boldsymbol{\Sigma}_C$. Using Eq. 8 and properties of the trace, we obtain:

$$\text{tr}\left\{\boldsymbol{\Sigma}_C^{-1}\boldsymbol{\Sigma}_q\right\} = \text{tr}\left\{\begin{pmatrix} \boldsymbol{D}_1\boldsymbol{\Sigma}_1 & -\boldsymbol{D}_1\boldsymbol{C}^T\boldsymbol{\Sigma}_2 \\ -\boldsymbol{D}_2\boldsymbol{C}\boldsymbol{\Sigma}_1 & \boldsymbol{D}_2\boldsymbol{\Sigma}_2 \end{pmatrix}\right\} = \text{tr}\left\{\boldsymbol{D}_1\boldsymbol{\Sigma}_1\right\} + \text{tr}\left\{\boldsymbol{D}_2\boldsymbol{\Sigma}_2\right\}.$$

Additionally, as $\boldsymbol{\Sigma}_C$ is a block matrix, $|\boldsymbol{\Sigma}_C| = |\boldsymbol{I}_1 - \boldsymbol{C}^T\boldsymbol{C}| = 1/|\boldsymbol{D}_1| = 1/|\boldsymbol{D}_2|$ [20, p. 114]. Similarly. as $\boldsymbol{\Sigma}_q$ is a diagonal matrix we have $|\boldsymbol{\Sigma}_q| = |\boldsymbol{\Sigma}_1||\boldsymbol{\Sigma}_2|$. Writing $\boldsymbol{\mu}_i = \boldsymbol{\mu}_i(\boldsymbol{x}_i)$ and noting

that $\boldsymbol{\mu}_p = 0$, we obtain:

$$D_{KL}(q_{\phi_1}(\cdot|\boldsymbol{x}_1)q_{\phi_2}(\cdot|\boldsymbol{x}_2)\|p_{\boldsymbol{C}}) = \frac{1}{2}\left[\boldsymbol{\mu}_1^T\boldsymbol{D}_1\boldsymbol{\mu}_1 - \ln|\boldsymbol{\Sigma}_1| - n_1 + \mathrm{tr}\left\{\boldsymbol{D}_1\boldsymbol{\Sigma}_1\right\}\right] \tag{9}$$

$$+ \frac{1}{2}\left[\boldsymbol{\mu}_2^T\boldsymbol{D}_2\boldsymbol{\mu}_2 - \ln|\boldsymbol{\Sigma}_2| - n_2 + \mathrm{tr}\left\{\boldsymbol{D}_2\boldsymbol{\Sigma}_2\right\}\right]$$

$$- \frac{1}{2}\left[\ln|\boldsymbol{D}_1| + \boldsymbol{\mu}_1^T\boldsymbol{D}_1\boldsymbol{C}^T\boldsymbol{\mu}_2 + \boldsymbol{\mu}_2^T\boldsymbol{D}_2\boldsymbol{C}\boldsymbol{\mu}_1\right].$$

## 2.6 Matrix parameterization

The matrix $\boldsymbol{C}$ is unknown and therefore must be either chosen apriori or be optimised. In our work, $\boldsymbol{C}$ is updated along with all other parameters at each update step, using the ELBO function as the loss function. By doing this, two challenges were faced and addressed. Firstly, $\boldsymbol{C}$ must be updated in such a way that $\Sigma_C$ is a valid covariance matrix. Secondly, a differentiable parameterization of $C$ is needed to allow for end-to-end learning. Conditions for validity of the update step are outlined in the following theorems.

**Theorem 2.1.** $\Sigma_C$ *defined as in Eq. 6 is positive semi-definite if and only if all singular values of $C$ are bounded by 1.*

*Proof.* Due to previous remarks, it is equivalent to show that $\boldsymbol{I}_1 - \boldsymbol{C}^T\boldsymbol{C}$ is positive semi-definite if and only if the singular values of $\boldsymbol{C}$ are bounded by 1. Further, this is equivalent to the showing that eigenvalues of $\boldsymbol{I}_1 - \boldsymbol{C}^T\boldsymbol{C}$ are non-negative if and only if the eigenvalues of $C$ are bounded by 1.

The eigenvalues of $\boldsymbol{C}^T\boldsymbol{C}$ are given by $\{\sigma_k^2(\boldsymbol{C})\}_{k=1}^{n_1}$ [21] meaning the eigenvalues of $-\boldsymbol{C}^T\boldsymbol{C}$ are given by $\{-\sigma_k^2(\boldsymbol{C})\}_{k=1}^{n_1}$. Applying Weyl's Theorem [19, p. 242] (stated in Appendix A.1.2) to $\boldsymbol{I}_1 - \boldsymbol{C}^T\boldsymbol{C}$ implies:

$$\lambda_1(\boldsymbol{I}_1) + \lambda_{n_1+1-k}(-\boldsymbol{C}^T\boldsymbol{C}) \leq \lambda_k(\boldsymbol{I}_1 - \boldsymbol{C}^T\boldsymbol{C}) \leq \lambda_{n_1}(\boldsymbol{I}_1) + \lambda_{n_1+1-k}(-\boldsymbol{C}^T\boldsymbol{C}) \tag{10}$$

$$\implies 1 \leq \lambda_k(\boldsymbol{I}_1 - \boldsymbol{C}^T\boldsymbol{C}) - \lambda_{n_1+1-k}(-\boldsymbol{C}^T\boldsymbol{C}) \leq 1$$

which combined with the relations above gives $\lambda_k(\boldsymbol{I}_1 - \boldsymbol{C}^T\boldsymbol{C}) = 1 + \lambda_{n_1+1-k}(-\boldsymbol{C}^T\boldsymbol{C}) = 1 - \sigma_{n_1+1-k}^2(\boldsymbol{C})$. Therefore all eigenvalues of $\boldsymbol{I}_1 - \boldsymbol{C}^T\boldsymbol{C}$ are non-negative if and only if all singular values of $C$ are bounded by 1. $\square$

If the inequality on $\sigma_1(\boldsymbol{C})$ is replaced by a strict inequality then $\boldsymbol{\Sigma}_C$ is guaranteed to be positive definite. This is clear as all eigenvalues of $\boldsymbol{I}_1 - \boldsymbol{C}^T\boldsymbol{C}$ are now positive which ensures positive definiteness. This guarantees the applicability of Eq. 9. A further restriction, outlined in Theorem 2.2, can be made which enables a different parameterization of $C$ which assumes a scaled orthogonal relationship between the views.

**Theorem 2.2.** *If $\boldsymbol{C} = \alpha\tilde{\boldsymbol{C}} \in \mathbb{R}^{n_2 \times n_1}$ where $|\alpha| \leq 1$ and $\tilde{\boldsymbol{C}}$ is a semi-orthogonal matrix ($\tilde{\boldsymbol{C}}^T\tilde{\boldsymbol{C}} = \boldsymbol{I}_1$) then $\boldsymbol{\Sigma}_C$ is positive semi-definite.*

*Proof.* Given the assumed structure on $\boldsymbol{C}$, we have the following:

$$\boldsymbol{I}_1 - \boldsymbol{C}^T\boldsymbol{C} = \boldsymbol{I}_1 - \alpha^2\tilde{\boldsymbol{C}}^T\tilde{\boldsymbol{C}} = (1 - \alpha^2)\boldsymbol{I}_1. \tag{11}$$

As $\boldsymbol{I}_1$ is positive semi definite, if $|\alpha| \leq 1$ then, so too is $\boldsymbol{I}_1 - \boldsymbol{C}^T\boldsymbol{C}$. $\square$

Alternatively, this could be seen as a corollary to Theorem 2.1 - due to the semi-orthogonality of $\tilde{\boldsymbol{C}}$, all singular values of $\tilde{\boldsymbol{C}}$ are equal to 1 which means $\sigma_i(\boldsymbol{C}) = \alpha$ with $|\alpha| \leq 1$. Again, if the condition on $\alpha$ is replaced with a strict inequality ($|\alpha| < 1$)) this guarantees applicability of Eq. 9. A simplification of Eq. 9 assuming this orthogonality condition can be found in Appendix A.2.

To ensure applicability of Eq. 9, we can either require (a) $\boldsymbol{C} = \alpha\tilde{\boldsymbol{C}}$ with $\tilde{\boldsymbol{C}}$ an orthogonal matrix and $|\alpha| < 1$ or (b) $\sigma_1(\boldsymbol{C}) < 1$. The latter requires a matrix factorisation of $\boldsymbol{C}$ which includes singular values. The most obvious approach is to use a singular value decomposition (SVD) of

$C$ i.e. $C = USV^T$ where $U$ and $V$ are orthogonal matrices and $S$ is the matrix of singular values i.e. $S = \text{diag}(\sigma_k(C))$. These singular values can be parameterized by $\sigma_k(C) = (1 - \exp(-\sigma_k)/(1 + \exp(-\sigma_k))$ for $\sigma_k \in \mathbb{R}$. To fully parameterize $C$ with the singular value constraint, a parameterization of orthogonal matrices $U$ and $V$ is needed. Similarly, a parameterization of orthogonal $\tilde{C}$ is needed for option (a). The following assumes $n_1 = n_2 = n$.

As discussed by Shepard et al. [22], there are four popular parameterizations of orthogonal matrices. All methods in their preliminary form parameterize at most a subset of the orthogonal matrices (at maximum those with determinant +1 or -1). These parameterizations must therefore be extended to map to the entire orthogonal matrix space. As it is the only one-to-one mapping, the rational Cayley transform has been chosen for use within JPVAE. In its original form, the Cayley transform tells us that for all orthogonal matrices with no eigenvalues equal to 1, there exists a unique skew-symmetric matrix $A \in \mathbb{R}^{m \times m}$ such that $O = (I + A)(I - A)^{-1}$ [22]. To extend the parameterization to the full space of orthogonal matrices, an extra matrix consisting of 1s and $-1$s along the diagonal is needed [23, 24]. Let $J = \text{diag}([\mathbf{1}_{m-r}; -\mathbf{1}_r])$ where $r = \text{floor}(m/(1 + \exp(-s)))$ for $s \in \mathbb{R}$. Then $O = J(I + A)(I - A)^{-1}$ is a mapping onto the space of orthogonal matrices. $\tilde{C}$ is therefore parameterized in full by $(n - 1)^2/2 + 1$ parameters: $(\{a_{ij} \in \mathbb{R} : i < j \quad \text{for} \quad i, j \in \{1, \cdots, n\}\}, \{s \in \mathbb{R}\})$.

## 2.7 Imputation

Once the model has been learned on the training data, missing views can be imputed. Assume data is available for view $j$, but not view $i$, given $x_j$ we can impute the missing value of $x_i$. Data $x_j$ is fed into the encoder for view $j$, $E_{j\phi}$, and latent variables are sampled giving $z_j = a$. An estimate of $z_i$ can be obtained using the conditional distribution of $z_i$ given $z_j$. This is fed through decoder $i$, $D_{i\theta}$, to produce an estimate of $x_i$ given $x_j$, $\tilde{x}_{i|j}$.

The joint marginal of $z_1$ and $z_2$ is assumed to be a multivariate normal with mean $[\boldsymbol{\mu}_1; \boldsymbol{\mu}_2]$ and covariance matrix $\boldsymbol{\Sigma}$. The maximum likelihood estimator for the mean and covariance are obtained and denoted by $[\hat{\boldsymbol{\mu}}_1; \hat{\boldsymbol{\mu}}_2]$ and $\hat{\boldsymbol{\Sigma}}$ respectively. As any subset of variables from a multivariate normal conditioned on a known second subset of variables also follows a multivariate normal distribution, the conditional distribution can be found explicitly. For $i \neq j$, the distribution of $z_i$ given $z_j = a$ is observed is:

$$z_i | z_j = a \sim N\left(\hat{\boldsymbol{\mu}}_i + \hat{\boldsymbol{\Sigma}}_{ij}\hat{\boldsymbol{\Sigma}}_{jj}^{-1}(a - \hat{\boldsymbol{\mu}}_j), \hat{\boldsymbol{\Sigma}}_{ii} - \hat{\boldsymbol{\Sigma}}_{ij}\hat{\boldsymbol{\Sigma}}_{jj}^{-1}\hat{\boldsymbol{\Sigma}}_{ji}\right) \tag{12}$$

where $\hat{\boldsymbol{\Sigma}}_{kl} \in \mathbb{R}^{n_k \times n_l}$ is the submatrix of $\hat{\boldsymbol{\Sigma}}$ corresponding to the variables associated with view $k$ and view $l$. The conditional mean $\mathbb{E}(z_i | z_j = a) = \hat{\boldsymbol{\mu}}_i + \hat{\boldsymbol{\Sigma}}_{ij}\hat{\boldsymbol{\Sigma}}_{jj}^{-1}(a - \hat{\boldsymbol{\mu}}_j)$ is then used as an estimate of the latent variables in latent space $i$ and fed into $D_{i\theta}$ to obtain $\tilde{x}_{i|j}$, the imputed value of $x_i$ given $x_j$.

## 3 Numerical experiments

Through a series of experiments the performance of JPVAE is explored for both imputation purposes as well as for downstream analyses like classification. As JPVAE enables imputation of missing views ($\tilde{\mathbf{X}}_{i|j}$), this ability is investigated alongside reconstruction of views ($\tilde{\mathbf{X}}_i$).

A multi-view dataset was created from the binary version of the popular MNIST dataset, which consists of handwritten digits from 0 to 9 [25]. For each image, the top half was taken as view 1 and the bottom half was taken as view 2. This dataset of halved images is referred to as hvdMNIST. The hvdMNIST dataset has the desirable property of having a strong correlation between views. The dataset contains $50,000$ training images and $10,000$ test images. Experiments are repeated with 5 different random seeds and the average and standard deviation reported. Details on the model architecture and training details can be found in Appendices A.3 and A.4 respectively.

Two variants of JPVAE are explored, where in each one the validity of $\Sigma_C$ is enforced via different ways. The first variant imposes a bound on singular values (such that $\sigma_1(C) < 1$). The second variant enforces a scaled orthogonality where $CC^T = C^TC = \alpha^2 I$, with the value of $\alpha$ set as $0.95$. The two variants of JPVAE are compared with the $C = 0$ case, which corresponds to completely disjoint VAEs for each view.

Without explicitly learning a correlation structure, correlation is present between the two generated latent spaces. By incorporating a joint prior with a non-zero cross-correlation as presented in Eq. 6

into the loss function, JPVAE increases the correlation between views in the latent space as shown in Figure 4. Table 1 illustrates the improvement in the ability to reconstruct view 2 given view 1 and vice versa when correlation structure is learnt. Enforcing the orthogonality restriction improves the performance of JPVAE compared with simply applying the singular value bound.

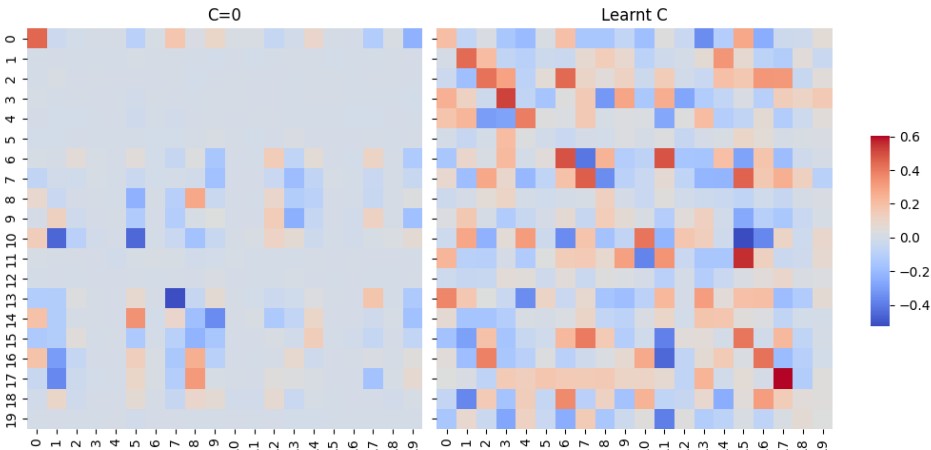

Figure 4: Empirical cross-correlation between view 1 and view 2 in the latent spaces. The left plot represents empirical cross-correlation for $C = 0$ and the right shows the same for $C$ learnt with the orthogonality restriction imposed. The Frobenius norm of these matrices are 1.703 and 3.448 respectively.

Table 1: Average reconstruction losses across the entire dataset. Best results in bold, standard deviation in brackets.

|  | Reconstruction | | Imputation | |
|---|---|---|---|---|
|  | $\tilde{X}_1$ | $\tilde{X}_2$ | $\tilde{X}_{1\|2}$ | $\tilde{X}_{2\|1}$ |
| $C = 0$ | 24.64 (0.37) | 25.56 (0.24) | 114.1 (2.3) | 127.5 (3.2) |
| $\sigma_1(C) < 1$ | 24.08 (0.44) | 25.02 (0.21) | 106.6 (1.4) | 117.4 (4.1) |
| $CC^T = C^TC = \alpha^2 I$ | **23.41** (0.29) | **23.98** (0.31) | **97.25** (1.9) | **106.6** (1.9) |

The improved performance of the method is enabled by the learnt correlation that prevents posterior collapse. Following the work by [26], the phenomenon of posterior collapse is indicated by the percentage of active units (AU). The activity of unit $u$ in the latent space is measured by $A_u = \text{Cov}_{\boldsymbol{x}}\left(\mathbb{E}_{u\sim q(u|\boldsymbol{x})}[u]\right)$. A unit is considered active, *i.e.* to not have suffered from posterior collapse, if $A_u \geq 10^{-2}$. A higher percentage of AUs are preserved when $C$ is learnt, with the best case scenario observed with the orthogonality constraint (Table 2). Wang et al. [12] proved that posterior collapse is equivalent to latent variable non-identifiability. This indicates that by enforcing the orthogonality restriction, we make the latent variable space identifiable.

Table 2: The percentage (%) of active units (AU) across latent spaces $(\boldsymbol{z}_1, \boldsymbol{z}_2)$, out of a total 40. Best results in bold, standard deviation in brackets.

|  | AU |
|---|---|
| $C = 0$ | 61 (1.4) |
| $\sigma_1(C) < 1$ | 66 (1.4) |
| $CC^T = C^TC = \alpha^2 I$ | **98.5** (2.2) |

The imputed views, $\tilde{X}_{1|2}$ and $\tilde{X}_{2|1}$, can be used for downstream tasks. If not all views are available for an individual, this allows techniques requiring all views to be applied. As an illustration of the performance of the imputed data in downstream tasks, a basic multi-layer perceptron classifier was

trained and tested on different combinations of reconstructed and imputed views. The performance of a classifier trained on the reconstructed data indicates that when $C$ is learnt, the relevant signal remains present in both the reconstructed and the imputed data. The performance on imputed data is greatly improved by learning $C$, and in particular by enforcing the orthogonality constraint (Table 3; results with standard deviation can be found in Figure 9.).

Table 3: Results for $(Y, Z)$ represent classification accuracy % for model trained on the training split of $Y$ and tested on the test split of $Z$. Accuracies for $(X_1, X_1)$ and $(X_2, X_2)$ with standard deviation in brackets are $93.59\%$ $(0.25)$ and $90.83\%$ $(0.23)$ respectively. Best results in bold. For clarity, results with standard deviation reported can be found in Figure 9.

| | View 1 | | | View 2 | | |
|---|---|---|---|---|---|---|
| | $(\tilde{X}_1, \tilde{X}_1)$ | $(\tilde{X}_1, \tilde{X}_{1\|2})$ | $(\tilde{X}_{1\|2}, \tilde{X}_{1\|2})$ | $(\tilde{X}_2, \tilde{X}_2)$ | $(\tilde{X}_2, \tilde{X}_{2\|1})$ | $(\tilde{X}_{2\|1}, \tilde{X}_{2\|1})$ |
| $C = 0$ | 89.07 | 47.21 | 77.22 | 85.88 | 51.83 | 78.81 |
| $\sigma_1(C) < 1$ | 89.67 | 64.50 | 80.22 | 86.46 | 67.06 | 82.38 |
| $CC^T = C^T C = \alpha^2 I$ | **90.31** | **77.22** | **83.54** | **87.80** | **76.55** | **86.30** |

Not only does learning correlation structure improve the ability to impute data, the reconstruction loss and classification accuracy for reconstructed data $x_i$ is improved compared with those scores seen when training each view on separate VAEs (Table 1). This may be due to the increased use of the latent space, as evidenced by the higher percentage of active units. Whilst a classifier trained on the imputed data from $C = 0$ demonstrates the retention of signal, the low accuracy seen for a classifier trained on reconstructed data and high reconstruction loss indicates that it does not retain the specific signature of the input. For example, taking the top half of a digit '2' as the input, it may correctly reconstruct the bottom half of a realisation of the digit '2' but not correctly reconstruct the specific realisation (as seen in Figure 1a).

Using the joint prior we see that the view with stronger signal (view 1) is bolstering the classification of the view with weaker signal (view 2). Whilst classification accuracy on $x_1$ is higher than that on $x_2$, the accuracy on the imputed data $\tilde{X}_{2\|1}$ sees a smaller drop, and is greater than that of $\tilde{X}_{1\|2}$. Notably, the accuracy on imputed data $(\tilde{X}_{2\|1}, \tilde{X}_{2\|1})$ is comparable to that on $(\tilde{X}_2, \tilde{X}_2)$ whilst the same for view 1 experiences a drop in performance.

## 4 Conclusions

A novel multi-view VAE approach has been proposed that natively strengthens the correlation between latent spaces via a joint prior. This is the first time that a connection between multi-view VAEs is made through a joint prior rather than a joint posterior, as has previously been implemented in the literature. Theoretical guarantees and parameterizations are presented that allow for end-to-end learning. By simultaneously preventing posterior collapse, JPVAE returns superior models and demonstrates a promising ability to impute missing data suitable for downstream tasks.

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

## A  Appendix / supplemental material

The Appendix contains additional results and figures to supplement the main body of text.

Theoretical results used within the manuscript are presented in Section A.1. This is followed by a simplification of the KL term when orthogonality constraints are assumed which is outlined in Section A.2. The model architecture and training details implemented in the numerical experiments are detailed in Sections A.4 and A.3 respectively. Lastly, additional results from the conducted numerical experiments can be found in Section A.5. All results throughout this manuscript are given to 4 significant figures, with standard deviations reported to 2 significant figures.

### A.1  Theoretical results

Theoretical results utilised within the paper are now presented.

### A.1.1  Block matrix inverse

The inverse of a block matrix, which is needed in Section 2.5 to determine the inverse of $\Sigma_C$ in order to derive the KL term, is now outlined.

For a matrix $\boldsymbol{A}$ with block matrix structure

$$\begin{pmatrix} \boldsymbol{A}_{11} & \boldsymbol{A}_{12} \\ \boldsymbol{A}_{21} & \boldsymbol{A}_{22} \end{pmatrix} \tag{13}$$

the following result holds [19, p. 18]:

**Lemma A.1.** *Assuming all relevant inverses exist, the inverse of $\boldsymbol{A}$ as defined in Eq. 13 is given by:*

$$\begin{pmatrix} (\boldsymbol{A}_{11} - \boldsymbol{A}_{12}\boldsymbol{A}_{22}^{-1}\boldsymbol{A}_{21})^{-1} & \boldsymbol{A}_{11}^{-1}\boldsymbol{A}_{12}(\boldsymbol{A}_{21}\boldsymbol{A}_{11}^{-1}\boldsymbol{A}_{12} - \boldsymbol{A}_{22})^{-1} \\ \boldsymbol{A}_{22}^{-1}\boldsymbol{A}_{21}(\boldsymbol{A}_{12}\boldsymbol{A}_{22}^{-1}\boldsymbol{A}_{21} - \boldsymbol{A}_{11})^{-1} & (\boldsymbol{A}_{22} - \boldsymbol{A}_{21}\boldsymbol{A}_{11}^{-1}\boldsymbol{A}_{12})^{-1} \end{pmatrix}. \tag{14}$$

Applying this with $\boldsymbol{A}_{ii} = \boldsymbol{I}_i$, $\boldsymbol{A}_{12} = \boldsymbol{C}^T$ and $\boldsymbol{A}_{21} = \boldsymbol{C}$ gives the result in the text.

### A.1.2 Weyl's inequality

The following inequality is used within the proof of Theorem 2.1 and concerns the sequence of eigenvalues of matrices. In contrast to the rest of the paper, the eigenvalues are indexed in non-increasing order, not non-increasing order of magnitude. For matrix $\boldsymbol{M}$ these are denoted by $\{\hat{\lambda}_j(\boldsymbol{M})\}_{j=1}^n$.

Weyl's inequality [19, p. 242, Corrollary 4.3.15] says:

**Lemma A.2.** *Let $\boldsymbol{A}, \boldsymbol{B} \in \mathbb{R}^{n \times n}$ be symmetric matrices. The following inequality holds*

$$\hat{\lambda}_j(\boldsymbol{A}) + \hat{\lambda}_1(\boldsymbol{B}) \le \hat{\lambda}_j(\boldsymbol{A} + \boldsymbol{B}) \le \hat{\lambda}_j(\boldsymbol{A}) + \hat{\lambda}_n(\boldsymbol{B}), \quad j = 1, \dots, n \tag{15}$$

Notice that $\lambda_k(\boldsymbol{I}_1) = \hat{\lambda}_j(\boldsymbol{I}_1) = 1$ for all $j, k$, and that as $-\boldsymbol{C}^T\boldsymbol{C}$ is negative semi-definite all eigenvalues are non-positive. This means $\hat{\lambda}_j(-\boldsymbol{C}^T\boldsymbol{C}) = \lambda_{n-j}(-\boldsymbol{C}^T\boldsymbol{C})$. Applying this lemma with $\boldsymbol{A} = -\boldsymbol{C}^T\boldsymbol{C}$ and $\boldsymbol{B} = \boldsymbol{I}_1$ gives Eq. 10.

## A.2 KL term simplification

Under the orthogonality assumption on $\boldsymbol{C}$, the KL term derived in Eq. 9 can be simplified. Specifically, if $\boldsymbol{C}\boldsymbol{C}^T = \boldsymbol{C}^T\boldsymbol{C} = \alpha^2\boldsymbol{I}$ for some $\alpha \in (0,1)$ then $\boldsymbol{D}_1 = \boldsymbol{D}_2 = \gamma\boldsymbol{I}$ where $\boldsymbol{I} = \boldsymbol{I}_1 = \boldsymbol{I}_2$ and $\gamma = 1/(1 - \alpha^2)$. Therefore Eq. 9 reduces to:

$$\begin{aligned} D_{KL}(q_{\phi_1}(\cdot|\boldsymbol{x}_1)q_{\phi_2}(\cdot|\boldsymbol{x}_1)||p_{\boldsymbol{C}}) = {} & \frac{1}{2}\left[\gamma\boldsymbol{\mu}_1^T\boldsymbol{\mu}_1 - \ln|\boldsymbol{\Sigma}_1| - n + \gamma\operatorname{tr}\{\boldsymbol{\Sigma}_1\}\right] \\ & + \frac{1}{2}\left[\gamma\boldsymbol{\mu}_2^T\boldsymbol{\mu}_2 - \ln|\boldsymbol{\Sigma}_2| - n + \gamma\operatorname{tr}\{\boldsymbol{\Sigma}_2\}\right] \\ & - \frac{1}{2}\left[n\ln|\gamma| + 2\gamma\boldsymbol{\mu}_1^T\boldsymbol{C}\boldsymbol{\mu}_2\right]. \end{aligned} \tag{16}$$

## A.3 Model architecture

All encoders and decoders for the JPVAE, as well as the neural network used for classification, consist of two dense layers with 512 units each. The latent distribution layers consists of two 20 unit dense layers which each parameterize the mean and the log of the variances of 20 normal random variables. The decoders output layer parameterizes the distribution of a Bernouilli random variable for each pixel. The classifier output layer parameterizes the categorical distribution with 10 outcomes .All activation functions were ReLu.

## A.4 Training details

The JPVAE used an Adam optimiser [27] with learning rate 0.001, trained for 30 epochs with a batch size of 32 and used binary cross entropy as the reconstruction error. The cyclical KL annealing schedule as introduced by [14] is implemented, with $M = 30$.

The classifier is trained for 50 epochs with a batch size of 32, step size of 0.01, except for on original data where it is trained for 15 epochs to prevent overfitting. Cross entropy loss is used as the loss function.

## A.5 Additional results

Additional results are presented in this section.

Figure 5 illustrates examples of reconstructing view 2 from view 1, with and without learning correlation natively, for a model trained with a different random seed to that displayed in Figure 1. Figures 6 and 7 illustrate examples of reconstructing view 1 from view 2, again with and without learning correlation natively, for models trained with different random seeds. As in Figure 1, the imputation of the missing view obtained when correlation of the latent spaces is learnt natively is of higher quality than when no correlation is learnt. To quantitively evaluate this superior performance, a simple multi-layer perceptron classifier is trained the concatenation of the reconstructed views. It is then tested on a combination of the reconstructed and imputed views. Explicitly, for datasets where *e.g.* view 2 is imputed from view 1, the classifier is trained on (the training split of) $[\tilde{X}_1; \tilde{X}_2]$ and tested on (the testing split of) $[\tilde{X}_1; \tilde{X}_{2|1}]$. Results of this classification task for various test datasets are displayed in Figure 8. Results for the classification task described in the main body of the text, and presented in Table 3, are illustrated in Figure 9 with standard deviation incorporated via error bars.

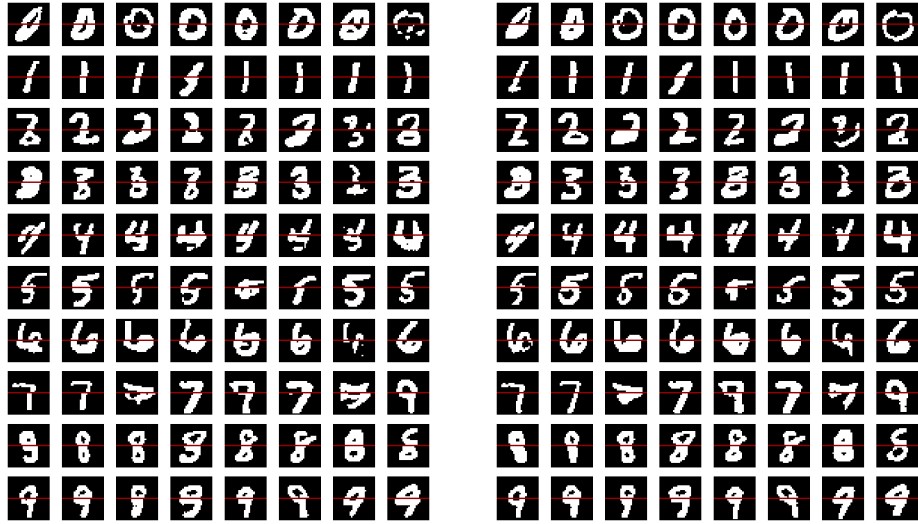

(a) No correlation learnt, but estimated empirically after training

(b) Correlation learnt natively

Figure 5: Additional realisation of an imputation of the bottom half of MNIST digits (view 2 of the data) using the top half of the image (view 1) on a JPVAE model trained with (a) independent priors (completely separate VAEs) and (b) a joint prior with learnt correlation structure between latent spaces. The cross entropy loss between true bottom half of image and imputation is 111.9 in (a) and 101.5 in (b).

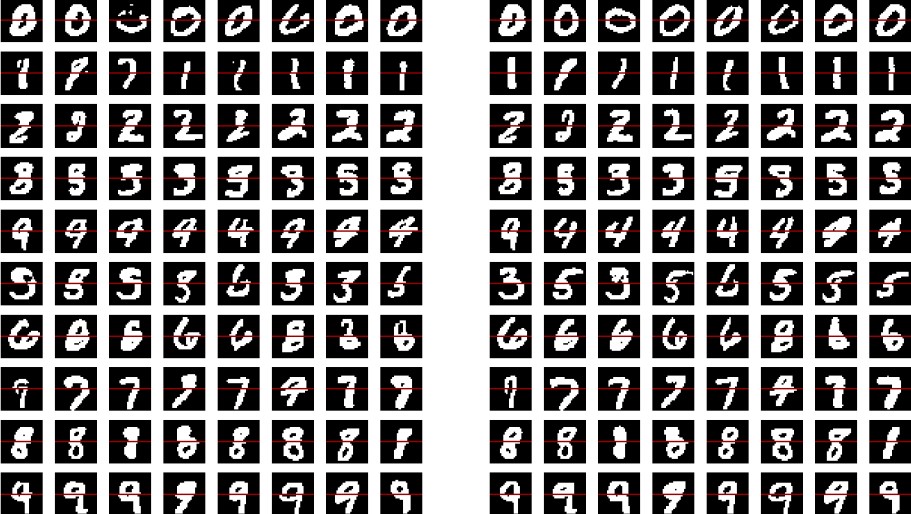

(a) No correlation learnt, but estimated empirically after training

(b) Correlation learnt natively

Figure 6: Imputation of the top half of MNIST digits (view 1 of the data) using the bottom half of the image (view 2) on a JPVAE model trained with (a) independent priors (completely separate VAEs) and (b) a joint prior with learnt correlation structure between latent spaces. The cross entropy loss between true top half of image and imputation is 117.8 in (a) and 100.2 in (b).

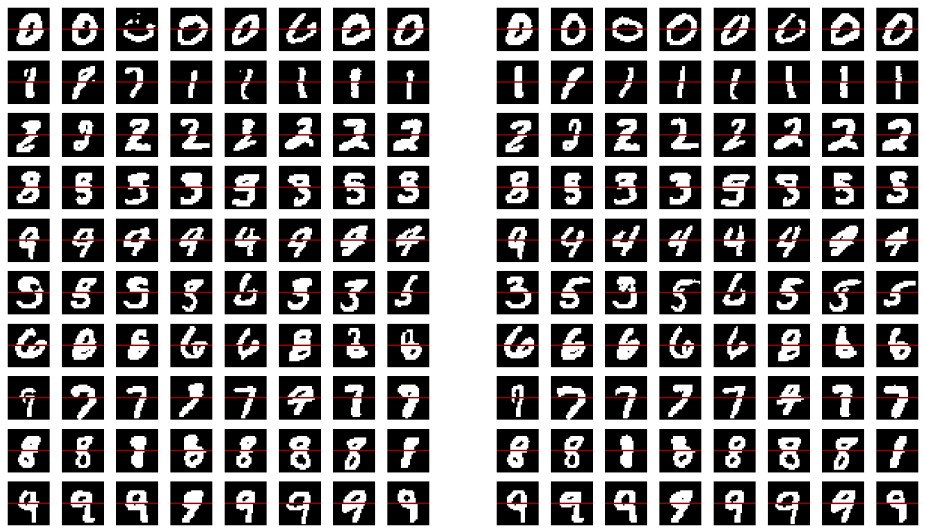

(a) No correlation learnt, but estimated empirically after training

(b) Correlation learnt natively

Figure 7: Additional realisation of an imputation of the top half of MNIST digits (view 1 of the data) using the bottom half of the image (view 2) on a JPVAE model trained with (a) independent priors (completely separate VAEs) and (b) a joint prior with learnt correlation structure between latent spaces. The cross entropy loss between true top half of image and imputation is 114.3 in (a) and 104.1 in (b).

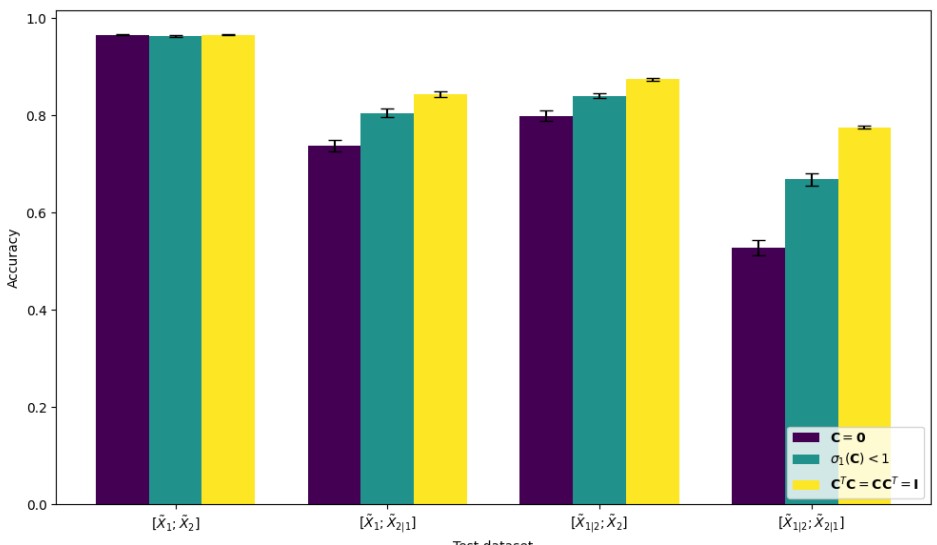

Figure 8: Results for $[\boldsymbol{Y}_1; \boldsymbol{Y}_2]$ represent classification accuracy % for model trained on the training split of $[\tilde{\boldsymbol{X}}_1; \tilde{\boldsymbol{X}}_2]$ and tested on the test split of $[\boldsymbol{Y}_1; \boldsymbol{Y}_2]$ (the column wise concatenation of $\boldsymbol{Y}_1$ and $\boldsymbol{Y}_2$). Accuracy for $[\boldsymbol{X}_1; \boldsymbol{X}_2]$ with standard deviation in brackets is $98.04\%$ $(0.074)$. Error bars present +/- one standard deviation.

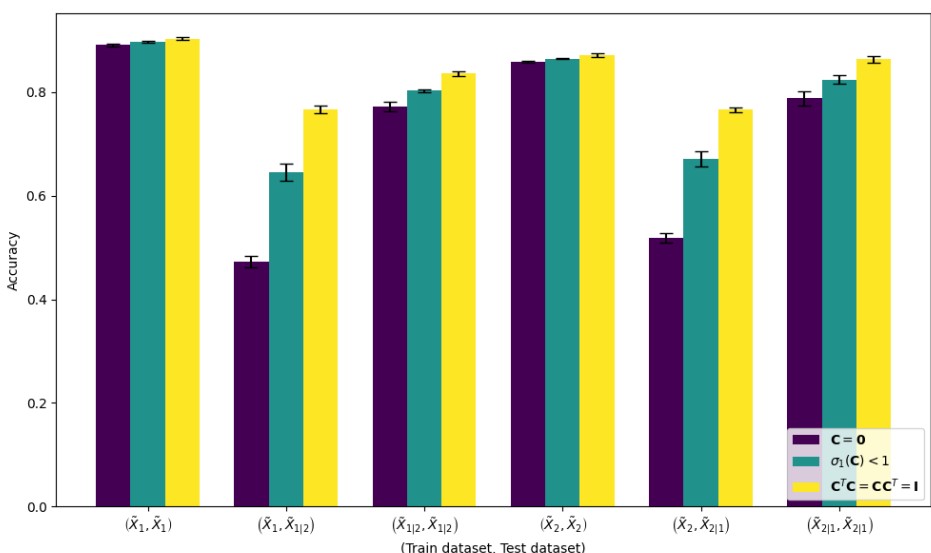

Figure 9: Results for $(\boldsymbol{Y}, \boldsymbol{Z})$ represent classification accuracy % for model trained on the training split of $\boldsymbol{Y}$ and tested on the test split of $\boldsymbol{Z}$. Accuracies for $(\boldsymbol{X}_1, \boldsymbol{X}_1)$ and $(\boldsymbol{X}_2, \boldsymbol{X}_2)$ with standard deviation in brackets are $93.59\%$ $(0.25)$ and $90.83\%$ $(0.23)$ respectively. Error bars present +/- one standard deviation.

