# OpenReview forum: "Correlating Variational Autoencoders Natively For Multi-View Imputation"
_NeurIPS.cc/2024/Workshop/UniReps — UniReps_

### Official Review · Reviewer_24G9 · 2024-10-05
**The paper is well written, background, theoretical part and method is very clear and concise. The scope fits very well to the workshop. Few questions regarding the results though.**

**Rating:** 7
**Confidence:** 2

**Review:**

There is very little to add to the introduction and the background.

1) I am missing a theoretical link to other alignment loss functions, as the proposed covariance structure shows similarities to other self-supervised loss functions (e.g. contrastive loss, barlow-twins...).

2) As a qualitative evaluation of Figure 1, a classifier could be trained on concatenated native views vs correlated views.

3) Does the correlation increase if one trains a single decoder with shared parameters for $z_1$ and $z_2$, instead of two separate decoders?

---

> ### Author Response · Authors · 2024-11-06
> **Review reply**
>
> Many thanks for your positive comments and your time taken in reviewing our work. In response to your questions:
> 1. Thank you for pointing this out,  a comment has been added to the introduction mentioning the similarity to self-supervised learning techniques such as Barlow twins. However, as our method is developed for multi-view data rather than noisy versions of the same view, no further discussed is had.
> 2. This has been implemented with detailed results also contained in the appendix. The results show that classification performed on the concatenation of the two views is greatly improved by enforcing correlation between the latent spaces.
> 3. As implementing a single decoder corresponds to a very different model structure and the implementation of a joint posterior (which is the focus of entire methods) this is beyond the scope of this work. However, this would be a worthwhile comparison, and we plan to compare the approach with other popular multi-view VAE methods (including those that implement joint posteriors) in future work.

---

### Official Review · Reviewer_gDcC · 2024-10-06
**Useful extension of the VAE framework for use in data from 2 views**

**Rating:** 6
**Confidence:** 3

**Review:**

In this paper the authors present a method to use VAEs in order to generate data coming from two distinct views, an example being medical data where the same patient is scanned by different scanners or under different conditions. Their method presents a way to train two VAEs such that the latent space distribution between both views are correlated, and through experiments demonstrate improvements in reconstruction over using two uncorrelated VAEs. The authors also demonstrated that by using the VAE structure, they are able to impute a missing view given the first one.

While the paper presents good ablation comparisons and contributes to the workshop, I believe that the paper needs to be compared to a model that performs the same task but is part of another family, rather than a special case of the same model.

---

> ### Author Response · Authors · 2024-11-06
> **Review reply**
>
> Many thanks for your time in reviewing our work. We agree it would be helpful to compare the model to a method from another family. This manuscript has focussed on the presentation of this novel method and any comparisons with other methods will be the focus of future work.

---

### Official Review · Reviewer_eahU · 2024-10-06
**A good idea of learning joint priors that model the correlation between latent spaces in the VAEs**

**Rating:** 7
**Confidence:** 4

**Review:**

The paper proposes an end-to-end training framework for learning joint priors that captures the correlations between latent spaces of multi-view data in VAEs.

The idea is described clearly, however it lacks sufficient results and visuals that shows that the idea generally works as expected. Moreover, the experiments were conducted on MNIST dataset, which is pretty simple and non-comprehensive. The authors also do not show enough visual results from the VAE multi-view imputation.

Overall, the idea is good, but it needs to be analyzed on more complex datasets, which can be the scope of a future publication.

---

> ### Author Response · Authors · 2024-11-06
> **Review reply**
>
> Many thanks for your time in reviewing our work. In response to your concerns regarding:
> - Insufficient results:  we have now performed repeats and applied the classifier to the image re-stitched together to further illustrate successful reconstruction/imputation.
> - Insufficient visuals: we have added to the appendix a figure illustrating imputing the top half of the image from the bottom, as well as replicates.
> - Lack of complexity on analysed dataset – we agree, MNIST has been used here to illustrate proof of concept and it is our plan to analyse its performance on more complicated data in future work.

---

### Decision · Program_Chairs · 2024-10-10

**Decision:**

Accept

**Comment:**

In light of the positive reviewers' feedback and relevancy of the submission, we are pleased to accept this paper for presentation at UniReps 2024. We kindly ask the authors to incorporate the reviewers' suggestions and feedback in the final camera-ready version of the manuscript.